# In Silico Analysis and In Vitro Characterization of the Bioactive Profile of Three Novel Peptides Identified from 19 kDa α-Zein Sequences of Maize

**DOI:** 10.3390/molecules25225405

**Published:** 2020-11-19

**Authors:** Jorge L. Díaz-Gómez, Ines Neundorf, Laura-Margarita López-Castillo, Fabiola Castorena-Torres, Sergio O. Serna-Saldívar, Silverio García-Lara

**Affiliations:** 1Escuela de Ingeniería y Ciencias, Tecnologico de Monterrey, 64849 Nuevo León, Mexico; cratker@gmail.com (J.L.D.-G.); lmlopez@tec.mx (L.-M.L.-C.); sserna@tec.mx (S.O.S.-S.); 2Department für Chemie, Institut für Biochemie, Universität zu Köln, D-50674 Köln, Germany; ines.neundorf@uni-koeln.de; 3Escuela de Medicina, Tecnologico de Monterrey, 64710 Nuevo León, Mexico; fcastorena@tec.mx

**Keywords:** anti-cancer, peptide, cell-penetrating, α-zein, antioxidant, ACE inhibitor, in silico

## Abstract

In this study, we characterized three novel peptides derived from the 19 kDa α-zein, and determined their bioactive profile in vitro and developed a structural model in silico. The peptides, 19ZP1, 19ZP2 and 19ZP3, formed α-helical structures and had positive and negative electrostatic potential surfaces (range of −1 to +1). According to the in silico algorithms, the peptides displayed low probabilities for cytotoxicity (≤0.05%), cell penetration (10–33%) and antioxidant activities (9–12.5%). Instead, they displayed a 40% probability for angiotensin-converting enzyme (ACE) inhibitory activity. For in vitro characterization, peptides were synthesized by solid phase synthesis and tested accordingly. We assumed α-helical structures for 19ZP1 and 19ZP2 under hydrophobic conditions. The peptides displayed antioxidant activity and ACE-inhibitory activity, with 19ZP1 being the most active. Our results highlight that the 19 kDa α-zein sequences could be explored as a source of bioactive peptides, and indicate that in silico approaches are useful to predict peptide bioactivities, but more structural analysis is necessary to obtain more accurate data.

## 1. Introduction

Bioactive peptides have been an important subject of research in recent years due to their wide range of bioactivities [1] and have been proposed as alternative, preventive, and therapeutic agents against diverse diseases, including cancer [2]. In comparison to traditional treatments, bioactive peptides have displayed diverse benefits, including low toxicity, a high affinity for target cells, high specificity, low accumulation in tissues, fewer side effects, and high tissue penetration [3,4].

For example, one of the most studied features of bioactive peptides is their capacity to penetrate the cell, hence named cell-penetrating peptides (CPPs) [5]. This class of peptides can pass through cell membranes and exert their bioactivity inside the cell or act as carriers for other molecules, such as proteins, peptides, lipids, antibiotics, and drugs [6]. Antioxidant activity, a key property of bioactive peptides, is related to the etiology of diseases like cancer, inflammatory diseases and diabetes mellitus [7]. Antihypertensive effects have also been suggested, as some peptides can inhibit angiotensin-converting enzyme (ACE) [8]. ACE is one of the main enzymes of the renin–angiotensin–aldosterone system and is responsible for the cleavage of angiotensin I to angiotensin II, which has a key role in the pathogenesis of hypertension [9].

To date, four methods have been described to generate peptides, namely enzymatic hydrolysis, chemical hydrolysis, microbial fermentation, and peptide synthesis [10]. Solid-phase peptide synthesis (SPPS) is frequently used to synthesize novel peptides, where the *C*-terminus of an amino acid is coupled to a functional group of a solid polymer. The *N*-terminal amino group and the side chains of the peptide are also protected. Peptide elongation is performed by cleavage of the *N*-terminal protector and coupling of the next protected amino acid [11]. For de novo synthesis, in silico prediction has been a powerful tool. Molecular interactions described for certain structures could predict their function or interaction with specific molecules and receptors [10]. This methodology is supported by diverse tools, such as in silico digestion, homology-based searches in databases of known proteins and peptides, and prediction by molecular docking and structural alignments [12].

Cereals, including maize, are a rich source of bioactive peptides [13]. Several studies have described their potential as anti-cancer agents [14,15,16,17]. Maize zeins are the main storage protein in kernels, and have been proposed as a source of peptides with diverse bioactivities, such as antioxidant, anti-inflammatory, anti-cancer, and antihypertensive [4,18,19,20,21,22]. Regarding the anti-cancer activity of maize peptides, one study reported that peptide fractions obtained from corn gluten meal (CGM) exhibited cytotoxic and anti-tumor activity in a hepatocarcinoma model in vitro and in vivo, with evidence suggesting that the peptides showed immunomodulatory and pro-apoptotic activities [23]. Ortiz-Martinez et al. described the antiproliferative and pro-apoptotic activities of peptides in HepG2 cells [16]. Several in vitro studies have shown the antioxidant activities of the peptides using different techniques to generate bioactive compounds [21,24,25,26,27]. Recently, we have reported that zein peptides have important antioxidant, cytotoxic, and pro-apoptotic activities in hepatocarcinoma cells [28]. Furthermore, some maize-derived peptides have significant antihypertensive activity [29,30]. The majority of these peptides are <5 amino acids long and exert their effects by inhibiting ACE.

Even though the bioactive profile of maize peptides has been studied extensively, most studies have focused on peptides obtained from protein hydrolysis using enzymes or other chemical or physical treatments. However, no study has used in silico tools to predict and characterize bioactive maize peptides.

The aim of this study is to predict and analyze α-zein derived peptides using in silico tools and to evaluate their cytotoxicity, cell-penetrating ability and antioxidant and ACE inhibitory activities in vitro. These data will be useful in defining if the bioactivities of the peptides derived from prolamins, which have limited structural information annotated in the databases, could be predicted following this strategy. Moreover, key structural features could be identified for the prediction of bioactivity.

## 2. Results

### 2.1. Characteristics of the Synthesized Peptides and In Silico Prediction of Bioactivity

The names and analytical data of the six peptides used in this study are detailed in Table 1. Of the three peptides, 19ZP3 had a positive charge of +1, 19ZP2 had a negative charge of −1 and 19ZP1 had a neutral charge. The level of purity achieved for 19ZP1 and 19ZP3 peptides was ≥95%, and for 19ZP2 was >70%. For the cellular uptake assay, the peptides were labelled with 5(6)-carboxyfluorescein (CF) at the *N*-terminus. As seen in Table 1, all experimentally measured masses match the calculated masses. Purities achieved were >90%, however all peptides displayed high hydrophobicity, which was even higher for the CF-labelled variants (for more analytical details, refer to Appendix A). Functional prediction assumed a higher probability of cell-penetrating activity for 19ZP3 (0.33), whereas 19ZP1 and 19ZP2 showed quite low probabilities (0.098 and 0.11, respectively). The probabilities for anti-cancer activities for the three peptides were below 0.01. The antioxidant probability values were 0.08, 0.125 and 0.1 for 19ZP1, 19ZP2 and 19ZP3, respectively. Finally, predictions for the inhibitory activity of ACE were 0.48 for 19ZP1, 0.375 for 19ZP2 and 0.45 for 19ZP3. This data implied that the peptides might possibly act/show activity as ACE inhibitors.

### 2.2. Predicted Peptide Structural Characteristics and Peptide-Solvent Interactions

Figure 1 shows 3D models of the predicted structures of the three synthesized peptides and the polar interactions between their amino acids. The three peptides have α-helical structures; 19ZP1 and 19ZP3 have three and 19ZP2 has two α-helices. The initial, inter helix and final segments of the peptide structures are composed of random coils. This data is consistent with the circular dichroism (CD) data under hydrophobic conditions. Most polar interactions for the three peptides occurred in the inner part of the α-helices, suggesting that secondary structures formed in the peptides. In the right-hand side of Figure 1, the electrostatic potentials for the three peptides are shown. In all cases, negative and positive electrostatic potential areas are seen. For 19ZP1, F1, N2, Q3, L4, N8, S9 and Q14 formed positive electrostatic potentials, whereas Q17, L18, L19, L24 and A25 formed negative electrostatic potentials. Amino acids Q1, L2, A8, A9 and F10 contributed positive electrostatic potentials and A3, D4, Y20, L21, H22, A23 and M24 contributed negative electrostatic potentials in the 19ZP2 model. For 19ZP3, A1, Y2, L3, Q4, A5, N12 and R16 contributed positive electrostatic potential and L9, A19 and A20 contributed negative electrostatic potentials. All three peptides displayed at least one negative and one positive electrostatic potential surface, according to the scale used (−1 to +1), and these data matched with the calculated net charges of the peptides (see Table 1).

The peptide-solvent (buffer: 0.15 mol/L NaCl) interactions are shown in Figure 2. The original predicted structure of the peptides was observed during simulations at 37 °C. In fact, 20 and 15 amino acids of 19ZP1 and 19ZP2, and all amino acids of 19ZP3, were able to interact with solvent particles. These amino acid-solvent interactions allow possible associations that might facilitate the bioactivities of these peptides.

### 2.3. Peptide Structure

No α-helical structures were present in the samples diluted in phosphate-buffered saline (PBS) only when measured by CD. The plots resembled random coil structures (Figure 3a). By contrast, when peptides were diluted with 50% (*v*/*v*) trifluoroethanol in PBS, hints of α-helical like structures were detected. Positive bands were observed at 195 nm and negative bands were detected at 202 and 222 nm for 19ZP1 and 19ZP2 plots, respectively (Figure 3b). These data indicate that in hydrophobic conditions, the peptides begin to form α-helical structures.

### 2.4. Cytotoxicity

The cytotoxic activity (Figure 4) indicated that the peptides were not cytotoxic to human embryonic kidney 293 (HEK-293) and hepatocyte derived cellular carcinoma (Huh7) cell lines at the concentration range used. Therefore, the half maximal inhibitory concentration (IC_50_) or IC_90_ were not calculated for these cell lines. Our data suggested that the peptides cannot affect the proliferation of normal or cancer cell lines under the conditions tested in this study.

### 2.5. Cell Penetration of Peptides

The cell penetration activity of the peptides is shown in Figure 5. The peptide sC18 was used [35] as a reference and positive control for cell penetration. Compared to the positive control, the new peptide samples penetrated HEK-293 cells to a far lesser extent (1/9). Interestingly, the exerted effect was similar for all three peptides (Figure 5a). In addition, a similar trend was seen when Huh7 cells were exposed to the peptides. Also in this case, the novel peptides penetrated the cells to a significantly lesser extent compared to the sC18 peptide, although here, the uptake was generally lower as in HEK-293 cells. Interestingly, the sC18 peptide was also not that efficiently taken up in HuH7 cells. Intensity values were nearly 1/5 lower in HuH7 compared to HEK-293 cells (Figure 5b). In summary, our results indicated that the experimental peptides had only low cell-penetrating activity in both cell lines compared to standards.

### 2.6. Antioxidant Activity

The antioxidant activity of the synthesized peptides is shown in Figure 6. The peptide 19ZP2 (1349.36 µM Trolox equivalents (TE)/g peptide) showed the highest antioxidant activity of all peptides, with a 3.5- and 142.5-fold higher activity compared to 19ZP1 (9.47 µM TE/g of peptide) and 19ZP3 (382.34 µM TE/g peptide), respectively. These differences were statistically significant (*p* < 0.05). Our findings indicate that the antioxidant capacity was different and dependent on the physicochemical characteristics of each peptide.

### 2.7. Angiotensin-Converting Enzyme-Inhibitory Activity

All three peptides exhibited ACE inhibitory activity (Figure 7). Of these, 19ZP1 had the lowest IC_50_ value (14.19 µM) and 19ZP2 and 19ZP3 had higher values, at 174.43 and 202.04 µM, respectively (Table 2). The inhibitory activity of 19ZP1 was not statistically different from the positive control captopril; moreover, its dose-response behavior was similar. The IC_50_ value of captopril was 0.29 µM. The dose-response behaviors of both 19ZP1 and captopril displayed similar logistic growth curves, even though the range of dose needed for each case was different. The dose-response curves of 19ZP2 and 19ZP3 displayed linear growth and did not reach 100% ACE inhibition in the range of concentrations used. Thus, 19ZP1 was the most effective in inhibiting ACE and the closest to captopril in efficacy.

## 3. Discussion

### 3.1. Peptide Properties

The peptides, 19ZP1, 19ZP2 and 19ZP3 were synthesized at different levels of purity (Table 1), which is peculiar in peptide synthesis and can be explained by the highly hydrophobic (≥20 amino acids) nature of these peptides. Large, hydrophobic peptides have lower yields and purity [11]. Only two peptides had a net positive charge of +1. A high positive charge increases the probability of the peptide to interact with the cell membrane of cancer cells [36]. Nevertheless, the estimated positive charge of our peptides is not high enough for a charge interaction. Our data suggest that 19ZP1 and 19ZP2 form α-helix structures under hydrophobic conditions. This behavior could be due to the use of dimethyl sulfoxide (DMSO) to solubilize the samples under high hydrophobicity. Peptide hydrophobicity can also explain why no α-helix structures were observed when PBS was used as the solvent. Peptides can change their conformation when their hydrophobic residues interact with the cell membrane, thereby facilitating cell penetration [37]. An example of this behavior is given by transportan 10 (TP10), a peptide which forms a hydrophobic random coil in its *N*-terminal region when it is embedded in the cell membrane, and an α-helical structure in its *C*-terminal region with positive charged Lys residues pointing towards the aqueous regions and the hydrophobic residues facing the inner side of the membrane bilayer [38].

### 3.2. In Silico Analysis

The 3D structure prediction of two of the three synthesized peptides revealed that there are hints, although no final formation, of α-helical structures, which is consistent with the curves obtained. The positive bands observed in the 195 nm section and the negative bands observed in the 202 and 222 nm sections of the CD plots for 19ZP1 and 19ZP2 confirm our findings (Figure 3b). However, peptide structure prediction does not consider hydrophobicity or other solvent-interaction characteristics obtained during molecular dynamics simulations. Nevertheless, in silico structure prediction results are in accordance with the secondary structure reported previously for α-zein, specifically the 19 kDa fraction [39]. The authors used molecular mechanics and dynamics simulations to reveal that 19 kDa α-zein has α-helical structures, formed by four residues per turn and non-polar residue chains forming a hydrophobic face, which form a triple superhelix and nine antiparallel helical segments [40]. In another study, 19 kDa α-zein was shown to form helical structures at pH > 6.8, which was in the range tested in this study [41]. Thus, the synthesized peptides partially retained the secondary structure of their native proteins. Based on electrostatic potential surface prediction and the −1 to +1 scale, all three synthesized peptides presented a positive and a negative electrostatic potential surface. Positive electrostatic potential surfaces increase peptide interaction with negatively charged cell membranes, especially in cancer cells, making it a priority in designing CPPs [42]. CPPs are composed of cationic residues like arginine, histidine and lysine, which are important for cellular uptake [43,44]. In this study, 19ZP2 contained only one histidine residue, whereas 19ZP3 had only one arginine in its primary structure. The low charge levels observed were not strong enough for interaction with cell membranes. The charge levels of CPPs have been reported to be higher (>10) [42,45,46]. Regarding peptide–solvent interactions, most amino acid residues were predicted to interact with PBS (pH 7.4) at 37 °C. This suggests that amino acid sequences are involved in cytotoxic, cell-penetrating and antioxidant activities. However, the peptides showed significant antioxidant properties, likely due to the presence of hydrophobic amino acids, which can scavenge radicals [47]. Proline was reported to scavenge the hydroxyl radical [48]. Interestingly, proline constituted part of the three synthesized peptides [49]. The 19ZP2 peptide contains methionine and histidine, which is able to chelate and reacts with free radicals, respectively [49,50]. Moreover, the aromatic amino acids, such as phenylalanine and tyrosine, may increase the antioxidant potential of some peptides [51]. Further studies are needed to evaluate the effect of pH and other conditions on the secondary structure of zein derived peptides for future applications.

### 3.3. Peptide Cytotoxicity and Cell-Penetrating Capacity

None of the peptides were cytotoxic at all concentrations tested in both cell lines. Li et al. reported cytotoxic effects of CGM-derived peptides in HepG2 cells with doses varying from 80 to 5120 µg/mL [23]. Peptides obtained from maize globulins showed cytotoxic activity in HepG2 cells at doses between 5 and 150 µg/mL [16]. We have previously reported the cytotoxic activity of zein hydrolysates obtained from four different *Zea mays* species with a range of IC_50_ of 1.16 to 1.78 µg/mL in HepG2 cells [28]. However, in these studies, purified peptides were not used. To our knowledge, this is the first time that 19 kDa α-zein was used to predict the bioactivity of peptides in silico. The peptide sequences used in the present study contained the motif QQLLPF, which as mentioned in the method sections, encompasses different reported bioactive peptides. This motif, together with the variability of the rest of the peptide sequences, was expected to induce 3D structural formations and increase their bioactive potential for cytotoxic and cell-penetrating activities. Although no cytotoxic effect was observed on the cell lines used, further studies are necessary to evaluate the potential anti-cancer activity of these peptides.

The three test peptides did not penetrate HEK-293 and Huh7 cells considerably compared to the control. Peptide hydrophobicity can explain why these peptides were not cytotoxic nor cell penetrating. The α-helical structures and amphipathic characteristics of peptides are important in their interaction with cell membranes [5]. Most CPPs studied have a hydrophilic and hydrophobic domain [6]; however, the peptides studied here are highly hydrophobic. Hydrophobic or negatively charged peptides have been scarcely reported [52]. CPPs penetrate cells by direct penetration, endocytosis or translocation through a transitory structure [5,53]. For direct penetration, the α-helical structure is important for forming a pore in the cell membrane; however, the peptide must have a strong positive charge to interact with the negatively charged cell membrane [53]. Here, 19ZP1, 19ZP2 and 19ZP3 have a low positive charge. Another possible reason for the failure to penetrate cells might be that α-helical structures did not form, as seen in CD analysis. There is no evidence of the presence of helical structures in aqueous conditions (PBS), and helix formation was only observed under hydrophobic conditions. Although hydrophobicity of the cell membrane could cause a shift in the structure, perhaps the interaction is not strong enough to cause it. For the endocytosis pathway, cell membrane-peptide recognition must occur so the peptide can be engulfed by the cell [54]; however, this mechanism is unlikely for 19ZP1, 19ZP2 and 19ZP3. Thus, the peptides may not have interacted with the membrane because their sequences were not recognized by the cell membrane proteins. Furthermore, the positive charge of some of the peptides were not strong enough to cause an interaction with the negative charge of the cell membrane, as suggested by the in silico analyses.

### 3.4. Peptide Antioxidant Activity

Peptides 19ZP1, 19ZP2 and 19ZP3 had different levels of antioxidant activity, ranging from 9.47 to 1349.36 µM TE/g peptide. This data is consistent with previous studies of maize-derived peptides. Zhou et al. reported 65.6 to 191.4 µM TE/g peptide antioxidant activities of 12 fractions obtained from three enzymatic treatments [55]. The peptides tested in this study had similar or higher antioxidant activities. Other studies have reported higher levels of activity of maize-derived peptides. Ortiz-Martinez et al. reported that the antioxidant activities of eight hydrolysate fractions obtained from two maize accessions were more than 1 mM TE/mg peptide [56]. For zein derived peptides, the antioxidant activities ranged from 1.42 to 1.57 mM TE/mg peptide. These findings are consistent with those of Díaz-Gómez et al., who reported similar antioxidant levels for zein derived hydrolysates (0.72 to 1.06 mM TE/g peptide) [27,28]. Another study reported that the antioxidant activities of hydrolysate fractions obtained from CGM ranged from 0.24 to 0.56 mM TE/mg peptide [57]. All these studies evaluated the antioxidant activity of maize peptides using the oxygen radical absorbance capacity (ORAC) assay. These studies also used hydroxyl, 2,2’-azino-bis(3-ethylbenzothiazoline-6-sulfonic acid ABTS and 2,2-diphenyl-1-picrylhydrazyl DPPH radical-scavenging activity assays, an Fe (II)-chelating ability assay, and the measurement of intracellular reactive oxygen species levels to evaluate the antioxidant activities of maize peptides [24,26,58,59]. Thus, the peptides synthesized in this study have antioxidant activity, but further studies regarding the mechanisms are needed to have a better understanding of the differences between extracted and synthesized maize peptides.

### 3.5. Peptide ACE Inhibitory Activity

Although all three peptides inhibited ACE, only 19ZP1 had a dose-response behavior similar to the positive control (captopril). The extent of ACE inhibition was similar to maize peptides reported in the literature. For example, an Ala-Tyr dipeptide obtained from hydrolyzed CGM had an IC_50_ of 14.2 µM, which is similar to 19ZP1 [60]. Lin et al. reported an IC_50_ of 0.037 mg/mL for the Ala-Tyr dipeptide [61]. Parris et al. reported an IC_50_ range of 3.26 to 7.77 mg/mL for peptides derived from germ obtained as a by-product of wet- and dry-milled maize [62]. Using the same units, the IC_50_ values of our samples were 0.04, 0.47 and 0.45 mg/mL for 19ZP1, 19ZP2 and 19ZP3, respectively. Thus, 19ZP1 was more effective than 19ZP2 and 19ZP3, which exerted similar effects. The ACE inhibitory activities of Z19 α-zein derived peptides, with IC_50_ values ranging from 1.7 to 57 µM [63] were consistent with our findings. Notably, the peptides were derived from the same fraction of α-zein. Huang et al. reported an IC_50_ of 0.44, 0.29 and 1.27 mg/mL for CGM peptides <1, <3 and <5 kDa, respectively [18]. Thus, the efficacy of 19ZP2 and 19ZP3 was comparable to other reports and 19ZP1 was most effective. The ACE inhibitory activity of a Met, Ile/Leu, Pro, and Pro tetrapeptide, obtained from zein hydrolysis, had an IC_50_ of 0.07 mg/mL [22], which is similar to that reported for 19ZP1. Notably, 19ZP1 has three of the four amino acids (except Met), suggesting that the amino acid sequence of the peptide is crucial for the level of ACE inhibitory activity. A recent study reported CGM-derived peptides with ACE inhibitory activity similar to our samples. The authors reported that 30.72% of the peptides were between 1 and 3 kDa in size, which is in the molecular weight range of the peptides tested in this study [64]. Further studies are needed to determine how the sequence, number and type of amino acids and their molecular weights affect the ACE inhibitory activity of zein peptides. The in vivo effect of the peptides on systemic blood pressure and other potential medical applications will be studied to get a clearer picture about their potential use.

### 3.6. Coherence between In Silico and Experimental Bioactivity Data

According to the prediction for the possible bioactivities of the synthesized peptides, the anti-cancer activities were low, which is consistent with the observed absent cytotoxicity on the tested cell lines. Moreover, low cell-penetrating probabilities (9.8–33%) of the three peptides matched with the low levels of cellular uptake observed by flow cytometry. The BIOPEP-UWM database was used to determine the antioxidant and antihypertensive activities of the synthesized peptide samples. This web tool matches the sequences of the peptides submitted with the other peptides and proteins in the database and calculates the probability of bioactivity, but it does not consider parameters like secondary or tertiary structure and prediction of potential bioactivities could be, therefore, underestimated. For example, antioxidant activity prediction for the peptides used in this study was low (20%), but the experimental data indicated considerable antioxidant activity compared to other reports. Moreover, the ACE inhibitory activity prediction showed higher probability values for all samples (37.5–48%) that correlated with the bioactivity observed in the experimental data and corresponds with other antihypertensive peptides reported in the BIOPEP-UWM database [63,65,66,67,68,69]. Further research is needed to predict peptide bioactivity profiles by using other algorithms and methodologies. Our results suggest the need for considering secondary and tertiary structures, peptide–molecule interactions, and peptide charge for in silico prediction tools.

## 4. Materials and Methods

### 4.1. Reagents

All chemicals were purchased from VWR (Darmstadt, Germany), Sigma-Aldrich (Taufkirchen, Germany), Merck (Darmstadt, Germany) and Fluka (Taufkirchen, Germany) unless stated otherwise.

### 4.2. Peptide Design

The following parameters were considered for predicting peptides used in this study. First, for the formation of helical structures, α-zein was chosen because of its ability to form helical structures (up to 60%) [39]. Second, three peptides with masses between 2 and 3 kDa were selected, owing to the relevance of low molecular mass of biopeptides from maize [4,55,61,70,71], and their ability to form helical secondary structures similar to the native protein. Last, the peptides with the motif QQLLPF, which is conserved in 19 kDa α-zeins, were chosen. This sequence contains different small bioactive peptides with various bioactivities, such as antioxidant and antihypertensive, and it serves as a stimulant of glucose uptake [27,63,72,73]. The Clustal Omega algorithm was used to perform the alignments [74], while using Jalview version 2.11.1.0 (The Barton Group, University of Dundee, Dundee, UK) as graphical interface [75] (Appendix A). Using these criteria, three peptide sequences, including 19ZP1 (FNQLAALNSAAYLQQQQLLPFSQLA), 19ZP2 (QLADVSPAAFLTQQQLLPFYLHAM) and 19ZP3 (AYLQAQQLLPFNQLVRSPAA) were selected, each obtained from a different 19 kDa α-zein (Uniprot accessions: P24449, P06674 and P06678) [76].

### 4.3. In Silico Prediction of Peptide Bioactivities: Simulation of Structural Changes and Peptide–Solvent Interactions

Biochemical features of the synthesized peptides were calculated using Peptide Property Calculator (Innovagen AB, Sweden, available at https://pepcalc.com/). Cell-penetrating probability was predicted using the CPPpred server (Conway Institute of Biomolecular and Biomedical Research, University College Dublin, Ireland) [77]. The iACP tool (University of Electronic Science and Technology, Zhongshan City, China) was used to predict anti-cancer properties [78]. The prediction of bioactive functions was done with the BIOPEP-UWM database [79]. All web tools used the peptide sequences to determine specific structural and/or chemical characteristics for predicting bioactivities. The calculations were made only for sequences without a fluorescent label. Peptide sequences were used to generate 3D models using QUARK online Ab initio protein structure, an online algorithm (University of Michigan, Ann Arbor, MI, USA) [80,81]. Based on these models, in silico analyses were performed to determine the cause of the bioactivity. Visual Molecular Dynamics version 1.93 (University of Illinois at Urbana-Champaign, Urbana, IL, USA) was used to simulate peptide interactions with a physiological solvent (0.15 mol/L NaCl) at 27 °C and 37 °C during 10 ns, using equilibration and molecular dynamics simulation [82]. PyMOL molecular graphics system version 2.3.4 (Schrödinger, Inc., New York, NY, USA) was used to generate 3D models of the data obtained from the simulations. PyMOL tools were employed to analyze polar interactions between amino acids, electrostatic potential on the peptide surface and peptide–solvent interactions [83]. The electrostatic potential on the peptide surface was simulated using the APBS Adaptive Poisson–Boltzmann Solver plugin version 3.7 (Pacific Northwest National Laboratory, Columbia, WA, USA) [84].

### 4.4. Peptide Synthesis and Purification

All peptides were synthesized using the Rink amide resin through SPPS, following the fluorenylmethoxycarbonyl/tert-butyl Fmoc/tBu-strategy [42]. Nα-Fmoc protected amino acids were obtained from IRIS Biotech (Marktredwitz, Germany) on a Syro II peptide synthesizer (MultiSynTech, Witten, Germany). The peptides were synthesized as *C*-terminal amides. For the cell uptake assays, the peptides were labelled with 5(6)-carboxyfluorescein (CF) at the *N*-terminus. Labelling was done on the solid phase with protected side chains by using two equivalents of CF, 1-[bis(dimethylamino)methyl]-1H-1,2,3-triazole [4,5-*b*]pyridinium-3-oxide hexa-fluorophosphate (HATU), and *N*,*N*-diisopropylethylamine (DIPEA) for 2 h at room temperature with constant shaking. Peptide cleavage from the resin and removal of side chain protecting groups were achieved using a mixture of concentrated trifluoroacetic acid (TFA) and triisopropylsilane (TIS) (1:1, *v*/*v*) for 19ZP1 and 19ZP3 peptides and a mixture of thioanisole and 1,2-ethandithiole (1:3, *v*/*v*) for 19ZP2 peptide for 3 h at room temperature. Thereafter, the samples were precipitated and washed five times in cold diethyl ether, solubilized with H_2_O:*tert*-butanol (1:3) and lyophilized for 24 h. The peptides were then analyzed by RP-HPLC (Nucleodur 100-5 C18 ec, Macherey-Nagel, Düren, Germany) coupled with electrospray ionization/mass spectrometry (ESI/MS, Thermo Scientific LTQ-XL, Darmstadt, Germany). The mobile phase consisted of a linear gradient of acetonitrile (20% to 70%) in 0.1% aqueous formic acid for 15 min. Reverse phase high pressure liquid chromatography (RP-HPLC) was performed for peptide purification using the Hitachi Elite LaChrom (Tokyo, Japan) on a Nucleodur C18ec, 100-5 (Macherey-Nagel, Düren, Germany). The linear gradient used was 35–50% B in A (A = 0.1% TFA in water; B = 0.1% TFA in acetonitrile) for 30 min at a flow rate of 6 mL/min. Only 19ZP2 could be purified because the other peptides were insoluble. After purification, the peptides were freeze dried and stock solutions were prepared in DMSO for in vitro assays.

### 4.5. Circular Dichroism Spectroscopy

The secondary structure of the synthesized peptides was analyzed by circular dichroism (CD) [85]. The spectra were scanned from 180 nm to 260 nm using 0.5 nm intervals at 20 °C. The JASCO J-715 spectropolarimeter was used in an N_2_ atmosphere. The CD spectra were measured with a 1 mm thick quartz cuvette. The reading parameters used included sensitivity, 100 mdeg; continuous scan; scan speed, 50 nm/min; response time, 2 s; and bandwidth, 1.0 nm. The peptides were dissolved in DMSO and diluted in 10 mM PBS, pH 7.4 and 50% (*v*/*v*) trifluoroethanol in PBS. The final peptide concentration used was 5 µM due to the hydrophobicity and presence of DMSO.

### 4.6. Cell Culture and Cell Viability Assay

Human embryonic kidney cells (HEK-293) and human hepatocarcinoma (HuH7) cells were used for in vitro experiments. HEK-293 was used as the non-cancerous control. Both cell lines were cultured in sterile conditions following international guidelines [86]. At a confluency of approximately 90%, culture splitting was performed by using 0.5 mg/mL trypsin- ethylenediaminetetraacetic acid (EDTA) [36]. For cell-based assays, the peptides were dissolved in DMSO at 10 mM concentration and stored at −20 °C. For cytotoxicity testing, the resazurin-based cell viability assay was performed using HEK-293 and HuH7 cell lines [85]. The cells were treated with various concentrations of peptide samples in free-serum media for 24 h; 70% EtOH was used as a positive control and cells exposed only to media were used as a negative control. Fluorescence was read at 595 nm on the Tecan Infinite M200 plate reader (Männedorf, Canton Zurich, Switzerland). Cell viability was determined according to the fluorescence read, using non-treated cells as 100% and positive controls as 0% viability reference.

### 4.7. Cellular Uptake Assay

The cellular uptake of peptides was analyzed by flow cytometry [85]. Cells were seeded in 24-well plates in growth media. Seed densities of 1 × 10^6^ and 1.6 × 10^6^ cells/well were used for HEK-293 and HuH7 cells, respectively. The cells were grown to 80% confluency, treated with 10 µM labelled peptides in triplicate with free-serum media, incubated for 1 h at 37 °C, washed with PBS, trypsinized and resuspended in serum-free media (without phenol red). Cellular uptake analysis was performed using the Guava EasyCyte flow cytometer (Merck, Darmstadt, Germany). The GRN-B (525/30) channel was used and cellular autofluorescence was subtracted from all measurements.

### 4.8. Antioxidant Activity Assay

Antioxidant activity was determined using the oxygen radical absorbance capacity assay. Synthesized peptides were evaluated using a previously-described method [28]. Trolox (6-hydroxy-2,5,7,8-tetramethylchroman-2-carboxylic acid) was used as the standard (Sigma-Aldrich, St. Louis, MI, USA) with fluorescein (Sigma-Aldrich, Saint Louis, MI, USA). 2,20-Azobis (2-amidinopropane) dihydrochloride (Sigma-Aldrich, Saint Louis, MI, USA) was used to generate peroxyl radicals, and the fluorescence loss signal was monitored for 1 h using a microplate reader (Biotek Synergy HTX, Winooski, VT, USA). The absorbance of excitation and emission were set at 485 and 538 nm, respectively. The results were expressed as µmol of Trolox equivalents (TE) per gram of peptide.

### 4.9. ACE Inhibition Assay

ACE inhibitory activity was evaluated using the ACE Activity Assay test (Sigma-Aldrich CS0002, Saint Louis, MI, USA). This assay is based on the hydrolysis of angiotensin I by ACE to yield active angiotensin II. This test uses a synthetic fluorogenic peptide as the substrate, and the resulting fluorescence is proportional to ACE activity [87]. Briefly, all reagents were diluted in the assay buffer, according to the manufacturer’s instructions. A total of 10 µL of peptide sample and positive control (captopril PHR1307, Sigma-Aldrich, Saint Louis, MI, USA), diluted in assay buffer at five different concentrations, and negative control (assay buffer) were pipetted into the wells of a 96-well black opaque plate. Then, 40 µL of ACE, provided in the kit, was added to sample and control wells. Sample blanks consisted of 50 µL of assay buffer. A standard curve was prepared in 100 µL assay buffer per well ranging from 0 to 8 nmol for calculating enzymatic activity. Subsequently, 50 µL of fluorogenic substrate warmed to 37 °C was added to experimental, control and blank sample wells. The reaction was carried out at 37 °C and fluorescence was read every minute for 5 min in a microplate reader (Biotek Synergy HTX, Winooski, VT, USA). The absorbance of excitation and emission were set at 320 nm and 405 nm, respectively. All determinations were done in duplicate. Linear regressions were determined to calculate the enzymatic activity for each sample dose.

One unit is defined as the amount of enzyme that releases one nmol of fluorescent substrate in 1 min at 37 °C. The results were expressed as milliunits of enzyme (mU). Finally, the IC_50_ of each peptide sample and captopril was determined using regression analysis of the corresponding dose–response curves. IC_50_ was defined as the concentration of peptide or control necessary to decrease ACE activity by 50%. This value was expressed as µM.

### 4.10. Statistical Analysis

Data are expressed as the mean values ± standard deviation. Analysis of variance and statistical difference (Tukey’s test) were performed using Minitab version 19.1.1 (State College, PE, USA) and significant differences were set at *p* < 0.05. All determinations were done in triplicate unless stated otherwise.

## 5. Conclusions

In this study, bioactive peptides identified from three 19 kDa α-zein were successfully synthesized. In silico analyses allowed for the estimation of physicochemical characteristics and potential bioactivities. The peptides 19ZP1 and 19ZP2 showed some indications of α-helical structures under hydrophobic conditions. None of the synthesized peptides showed cell cytotoxicity and cell-penetrating activities in HEK-293 and HuH7 cells. Interestingly, all peptide samples presented significant in vitro antioxidant and ACE inhibitory activities. Some of the contrasting results observed between in silico and experimental results could be explained by the nature and limitations of the prediction tools. Further studies would be beneficial in determining the efficacy of simulation analyses for predicting bioactivity and evaluating the in vitro and in vivo bioactive profile of peptides. Evaluations of their in vivo antihypertensive activities and in vitro antioxidant studies would be appropriate to better define their potential pharmacological applications.

## Figures and Tables

**Figure 1 molecules-25-05405-f001:**
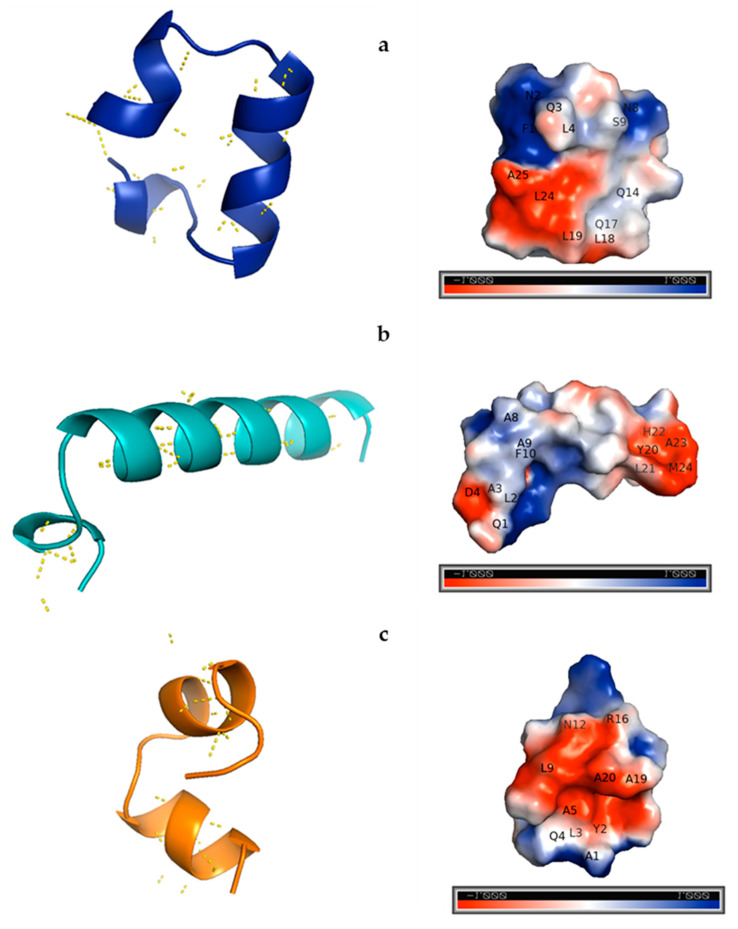
Left side: predicted 3D structure of the synthesized peptides and polar interactions between amino acids. Right side: electrostatic potential according to the predicted structures of the synthesized peptides. The QUARK online Ab initio protein structure algorithm was used to obtain the models and PyMOL molecular graphics system version 2.3.4 was used to generate the 3D model images. The Adaptive Poisson–Boltzmann Solver (APBS) plugin version 3.7 was used to calculate and predict the electrostatic potential surface. The electrostatic potential scale ranges from −1 to +1. Amino acids contributing with the potential are represented as one letter codes and their number in the sequence. (**a**) 19ZP1; (**b**) 19ZP2; (**c**) 19ZP3.

**Figure 2 molecules-25-05405-f002:**
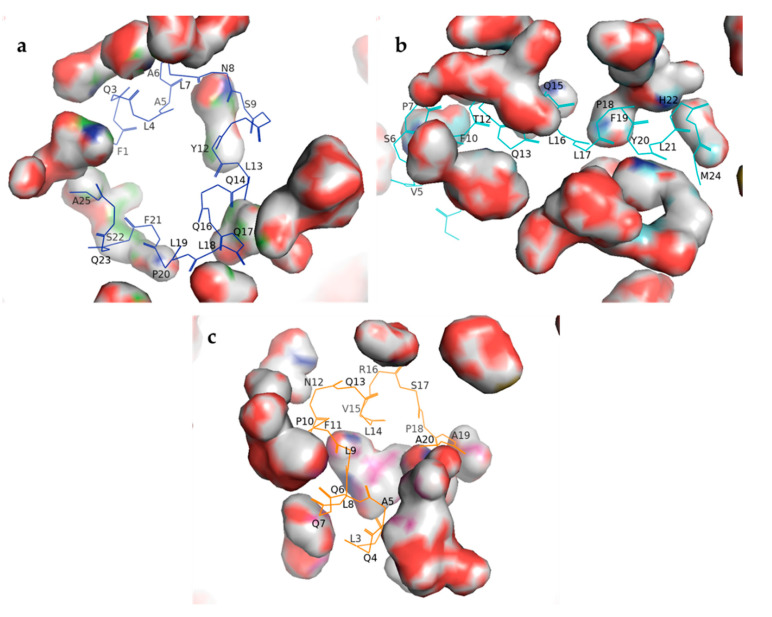
Peptide-solvent interactions based on the predicted structures of the synthesized peptides. The QUARK online Ab initio protein structure algorithm was used to obtain the models. Visual Molecular Dynamics version 1.93 was used to simulate peptide interactions with a physiological solvent (NaCl 0.15 mol/L, 37 °C and 10 ns) using equilibration and molecular dynamics simulations. PyMOL molecular graphics system version 2.3.4 was used to generate the 3D images. The surface of interaction is represented as solid bodies surrounding the peptide structure. Amino acids contributing to the peptide-solvent interaction are represented as one letter codes and their number in the sequence. (**a**) 19ZP1; (**b**) 19ZP2; (**c**) 19ZP3.

**Figure 3 molecules-25-05405-f003:**
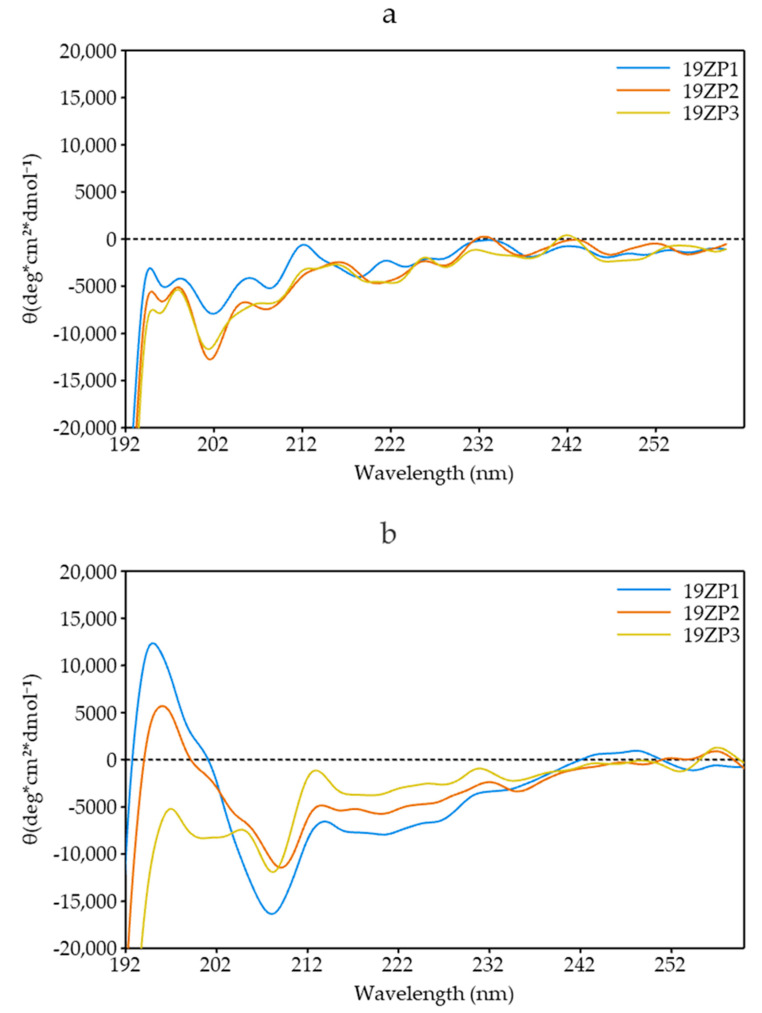
Structure of peptides analyzed by circular dichroism. The spectra were acquired at a peptide concentration of 5 µM in 10 mM phosphate buffer, pH 7.0 (**a**) and in 50% (*v*/*v*) trifluoroethanol in phosphate-buffered saline (PBS) (**b**).

**Figure 4 molecules-25-05405-f004:**
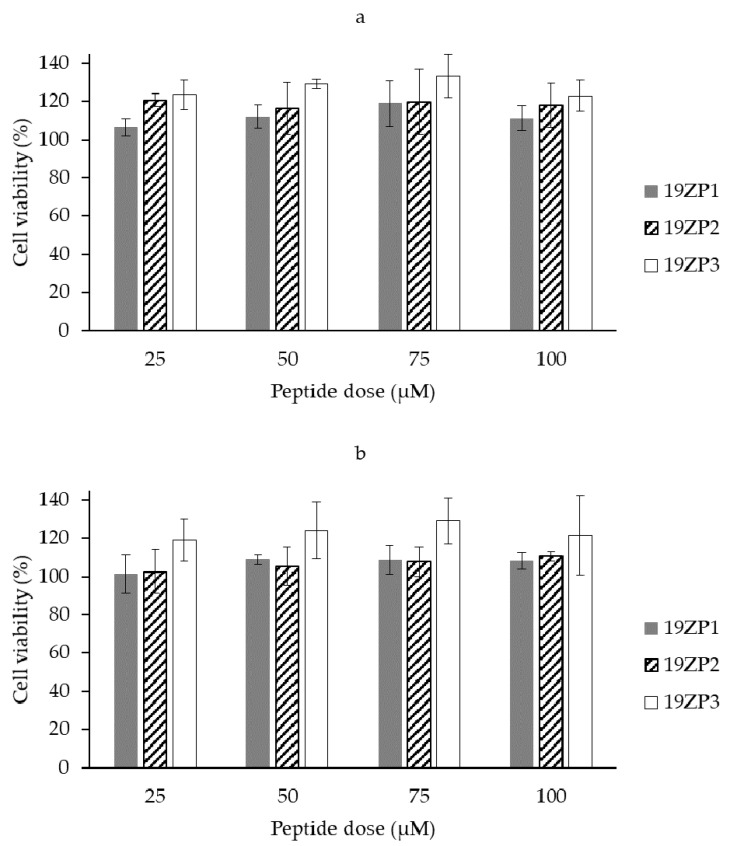
Cell viability of human embryonic kidney 293 HEK-293 (**a**) and hepatocyte derived cellular carcinoma Huh7 (**b**) after 24 h of peptide treatment measured by resazurin assay. Untreated cells were used as a negative control. Results are expressed as mean values ± standard deviation; *n* = 3. There were no statistical differences among samples according to Tukey’s test (*p* < 0.05).

**Figure 5 molecules-25-05405-f005:**
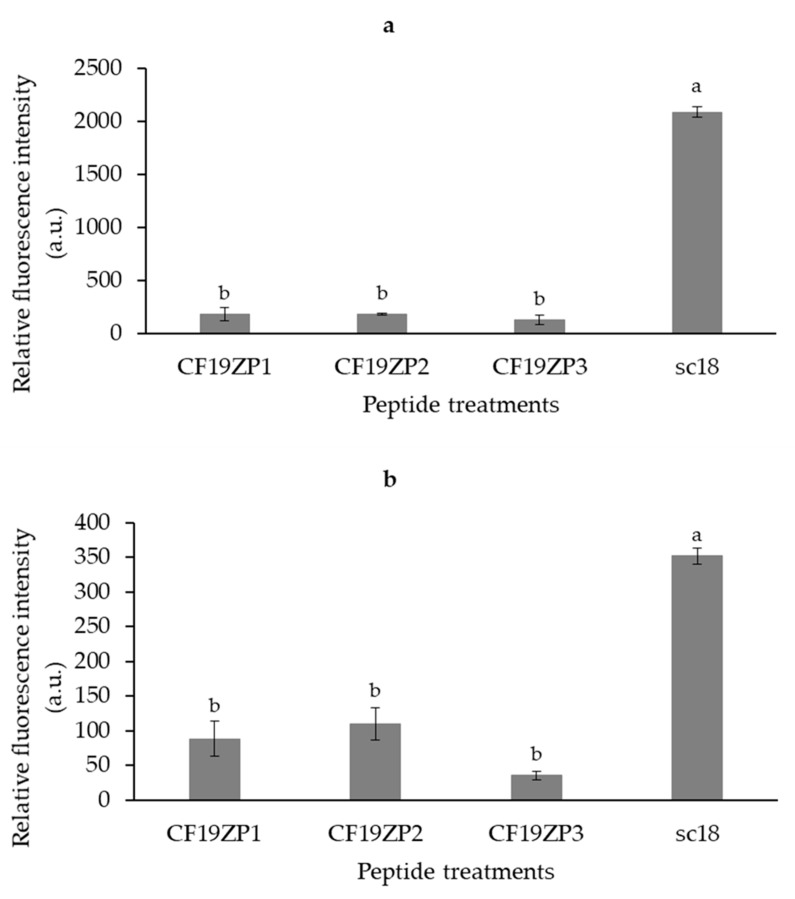
Cellular uptake of peptides by HEK-293 (**a**) and HuH7 (**b**) cells as measured by flow cytometry. The cells were treated with 10 µM peptide for 1 h. The peptide sC18 was used as a positive control. Results are expressed as mean values ± standard deviation; *n* = 3. Superscript letters represent statistical differences among samples. Statistical differences were determined by Tukey’s test (*p* < 0.05).

**Figure 6 molecules-25-05405-f006:**
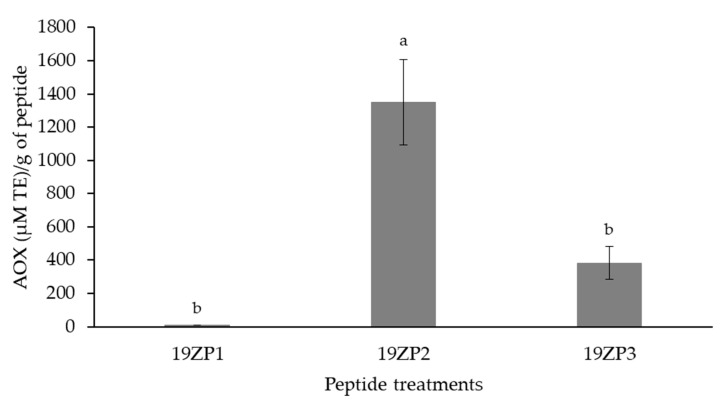
Evaluation of the antioxidant capacity (AOX) of the peptides using the oxygen radical absorbance capacity (ORAC) assay. Results are expressed as mean values ± standard deviation; *n* = 3. Superscript letters represent statistical differences among samples. Statistical difference was determined by Tukey’s test (*p* < 0.05). TE: Trolox equivalents

**Figure 7 molecules-25-05405-f007:**
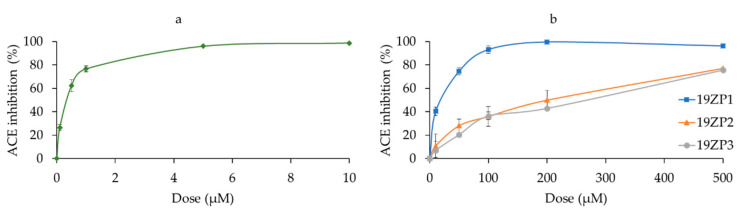
In vitro ACE inhibitory activity of peptides (**b**) compared to the positive control captopril (**a**). Dose-response curves were calculated using enzymatic activity values from the kinetic assay by reading fluorescence intensity (excitation 320 nm/emission 405 nm) every minute for 5 min for each dose. Results are expressed as values ± standard deviation; *n* = 2. Superscript letters represent statistical differences among samples. Statistical difference was determined by Tukey’s test (*p* < 0.05).

**Table 1 molecules-25-05405-t001:** Name and analytical data of the peptide sample used.

Name	Peptide Sequence	MW_calc_ [Da] ^1^	Mw_exp_ [Da]	Net Charge	Purity ^2^	CPP ^3^	ACP ^4^	AOP ^5^	ACEIP ^5^
19ZP1	FNQLAALNSAAYLQQQQLLPFSQLA	2777.14	2777.88		≥95%	0.098	0.000009	0.08	0.48
19ZP2	QLADVSPAAFLTQQQLLPFYLHAM	2702.13	2702.54	−1	≥70%	0.11	0.000004	0.125	0.375
19ZP3	AYLQAQQLLPFNQLVRSPAA	2227.56	2228.11	+1	≥95%	0.33	0.005109	0.1	0.45
CF19ZP1	CF-FNQLAALNSAAYLQQQQLLPFSQLA	3135.44	3136.07		>90%	nd	nd	nd	nd
CF19ZP2	CF-QLADVSPAAFLTQQQLLPFYLHAM	3060.43	3060.19	−1	>70%	nd	nd	nd	nd
CF19ZP3	CF-AYLQAQQLLPFNQLVRSPAA	2585.86	2586.32	+1	>90%	nd	nd	nd	nd

CF: 5(6) carboxyfluorescein; Da: daltons; MW_calc_: calculated mass; Mw_exp_: experimental mass; nd: data not determined for the labelled peptides. CPP: cell-penetrating probability; ACP: anticancer probability; AOP: antioxidant probability; ACEIP: angiotensin-converting enzyme inhibition probability. Using peptide sequencing, the following tools were used to predict the bioactive characteristics of the peptides: ^1^ Mass was calculated using the peptide property calculator [31]. Da: daltons. ^2^ Peptide purity was calculated using chromatographic spectra data obtained from synthesis, using reversed phase high-pressure liquid chromatography (RP-HPLC) coupled with electrospray ionization mass spectrometry (ESI/MS). ^3^ Cell penetrating probability was calculated using CPPpred server (Conway Institute of Biomolecular and Biomedical Research, University College Dublin, Ireland) [32], ^4^ Anticancer activity was predicted using iACP tool (University of Electronic Science and Technology, Zhongshan City, China) [33]. ^5^ Antioxidant and angiotensin-converting enzyme (ACE) inhibition probabilities were determined using BIOPEP-UWM database [34].

**Table 2 molecules-25-05405-t002:** Half maximal inhibitory concentrations (IC_50_) for ACE inhibition.

Sample	IC_50_ (µM)
Captopril	0.29 ± 0.02 ^b^
19ZP1	14.19 ± 0.12 ^b^
19ZP2	174.43 ± 49.6 ^a^
19ZP3	202.04 ± 0.63 ^a^

Results are expressed as values ± standard deviation; *n* = 2. Superscript letters represent statistical differences among samples. Statistical difference was determined by Tukey’s test (*p* < 0.05).

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
