# Peer review of "In Silico Analysis and In Vitro Characterization of the Bioactive Profile of Three Novel Peptides Identified from 19 kDa α-Zein Sequences of Maize"

_molecules, 2020, doi:10.3390/molecules25225405_

Round 1

Reviewer 1 Report

Dear Authors,
having reviewed and evaluated the work entitled: "In silico analysis and in vitro characterization of the bioactive profile of three novel peptides identified from the 19 kDa-zein sequences of maize", I evaluate the work well, with a few minor corrections below.

Suggestion 1. format table 1 to make it more readable
Suggestion 2. Line 80- 93. As example: “The probabilities for anti-cancer activities for the three peptides were below 0.01. The antioxidant probabilities values were 0.08, 0.125 and 0.1 for 90 19ZP1, 19ZP2 and 19ZP3, respectively. Finally, prediction for the inhibitory activity of ACE were 0.48 91 for 19ZP1, 0.375 for 19ZP2 and 0.45 for 19ZP3.” Probability of cell penetration? probability of bioactivity? I don't understand where this is coming from? Is it a statistical probability (I assume not - but it's not clear)? The probabilities discussed in section “2.1. Characterization of the synthesized peptides and the prediction of in silico bioactivity are not sufficiently elucidated. Do they result from the formulas of individual bioinformatics tools? If so, the formulas should be described in the "Methods" section.
Suggestion 3. Line 123-128: “The peptide-solvent (buffer: 0.15 mol/L NaCl) interactions are shown in Figure 2. No change in the original predicted structure of the peptides was observed during simulations at both temperatures (only 37°C simulations are shown). In fact, almost all amino acids of 19ZP1 and 19ZP2 and all amino acids of 19ZP3 were able to interact with solvent particles. These amino acid–solvent interactions might contribute to molecular mechanisms responsible of some bioactivities of the peptide samples” - too general sentence, not supported by any evidence from the conducted research. In the Discussion section, this topic was also not adequately explained. Please complete the appropriate part of the manuscript.
Suggestion 4. Line 376: BIOPEP-UWM database is correct!
Suggestion 5. As example line 314: CGM - this abbreviation has not been clarified.

Author Response

October 23th, 2020

Kyree Zhang
Assistant Editor

Journal name: Molecules

Manuscript ID: molecules-980474

Title: In silico analysis and in vitro characterisation of the bioactive profile of three novel peptides identified from 19 kDa α-zein sequences of maize

Dear Professor Zhang:

We keenly acknowledge the opportunity to improve our manuscript. We are submitting the revised manuscript following the suggestions from both reviewers. In consequence, we responded to all the comments point-by-point below. We are confident that the suggestions have been satisfactorily resolved. Text additions and modifications are highlighted (yellow) in the original draft

Yours sincerely

Professor Silverio García-Lara

-Reviewer 1

0. I evaluate the work well, with a few minor corrections below.

[Thank you very much for your encouragement!]

  1. Suggestion 1. Format table 1 to make it more readable

[Ok, table 1 has been modified to make it more readable reducing the names and the column format].

  1. Suggestion 2. Line 80- 93. As example: “The probabilities for anti-cancer activities for the three peptides were below 0.01. The antioxidant probabilities values were 0.08, 0.125 and 0.1 for 90 19ZP1, 19ZP2 and 19ZP3, respectively. Finally, prediction for the inhibitory activity of ACE were 0.48 91 for 19ZP1, 0.375 for 19ZP2 and 0.45 for 19ZP3.” Probability of cell penetration? Probability of bioactivity? I don't understand where this is coming from? Is it a statistical probability (I assume not - but it's not clear)? The probabilities discussed in section “2.1. Characterization of the synthesized peptides and the prediction of in silico bioactivity are not sufficiently elucidated. Do they result from the formulas of individual bioinformatics tools? If so, the formulas should be described in the "Methods" section.

[We acknowledge this observation. The probabilities reported in our study corresponded to the values displayed in each analysis performed using the different bioinformatics tools (algorithms) referenced in the manuscript. We will explain each value below:

  • The cell penetration probabilities were determined using CPPpred (mentioned at line 378). This tool uses a defined set of CPPs and non-CPPs and compares the submitted sequences to the datasets using a neural network to return the prediction of cell penetration for each submitted peptide. Reference: Holton, T. A.; Pollastri, G.; Shields, D. C.; Mooney, C. CPPpred: Prediction of cell penetrating peptides. Bioinformatics 2013, 29, 3094–3096, doi:10.1093/bioinformatics/btt518.
  • The antioxidant and ACE inhibitory probabilities were determined using the BIOPEP-UWM database; which compiles a large set of reported peptides with different bioactivities (mentioned at line 381). Then, in the “Analysis” menu, BIOPEP-UWM offers an option for the calculation of the profile of potential activity for proteins or peptides. This calculation takes into account different parameters, including number and type of amino acids, number of fragments in the sequence with a given activity, number of repetitions of bioactive fragments and the reported EC50 of those bioactive fragments if they are reported. Reference: Minkiewicz, P.; Iwaniak A.; Darewicz M. BIOPEP-UWM database of bioactive peptides: Current opportunities. J. Mol. Sci. 2019, 20, 5978, doi:10.3390/ijms20235978.
  • Probability of anticancer activity was estimated using the iACP tool (mentioned at line 379). In a similar manner than CPPred and BIOPEP-UWM, iACP uses a dataset of anticancer and non-anticancer peptides to perform the calculations. This algorithm utilizes a pseudo amino acid composition approach (Chou’s PseACC), a g-gap dipeptide composition and a SVM (support vector machine) classification algorithm to give the probability of the anticancer activity of the submitted peptides. Reference: Chen, W.; Ding, H.; Feng, P.; Lin, H.; Chou, K.-C. iACP: A sequence-based tool for identifying anticancer peptides. Oncotarget 2016, 7, 16895-16909, doi:10.18632/oncotarget.7815.

All the formulas used by the bioinformatics tools can be found in their corresponding references.]

  1. Suggestion 3. Line 123-128: “The peptide-solvent (buffer: 0.15 mol/L NaCl) interactions are shown in Figure 2. No change in the original predicted structure of the peptides was observed during simulations at both temperatures (only 37°C simulations are shown). In fact, almost all amino acids of 19ZP1 and 19ZP2 and all amino acids of 19ZP3 were able to interact with solvent particles. These amino acid–solvent interactions might contribute to molecular mechanisms responsible of some bioactivities of the peptide samples” - too general sentence, not supported by any evidence from the conducted research. In the Discussion section, this topic was also not adequately explained. Please complete the appropriate part of the manuscript.

[Thank you for this observation. These section was significantly modified as follows:

  • Results (2.2) at line 124…The peptide-solvent (buffer: 0.15 mol/L NaCl) interactions are shown in Figure 2. The original predicted structure of the peptides was observed during simulations at 37°C. In fact, 20 and 15 amino acids of 19ZP1 and 19ZP2, and all amino acids of 19ZP3 were able to interact with solvent particles. These amino acid–solvent interactions allow possible associations that might facilitate bioactivities of these peptides.
  • Discussion (3.2) at line 256…This suggests that amino acid sequences are involved in cytotoxic, cell-penetrating and antioxidant activities.]

    4. Suggestion 4. Line 376: BIOPEP-UWM database is correct!

[At line 381, the word has been changed accordingly].

  1. Suggestion 5. As example line 314: CGM - this abbreviation has not been clarified.

[The abbreviation has been clarified now at line 61 which is the first mention in the manuscript.]

Reviewer 2 Report

The MS "In In silico analysis and in vitro characterisation of the bioactive profile of three novel peptides identified from 19 kDa α-zein sequences of maize" represents, in my view, an interesting and representative paper to the understanding of bioactive peptides derived from maize. I have no concerns whatsoever on the experimental design nor in the results obtained by the researchers.

My main suggestion is to ask a professional Editing service to proofread the manuscript to remove typographical errors and/or loose, unstructured sentences. There are some questionable constructions (e.g. what does College represent to the version of Minitab? line 473; Suiza at line 430, which I assume the authors intended to mean Switzerland?). 

For instance, if one considers the abstract alone, a few suggestions that could be made are:

  • Please remove sentences from line 15 up to the word ‘scarce’ – line 17 on the abstract.
  • Substitute penetrating (line 22) to penetration
  • Remove ‘a’ from line 29

Apart from that, please remove the table from Figure 7, which is now bound to the same figure file. I suppose tables should not be sent as figures.

These suggestions do not take merit away from the paper, but indeed could help it gain some more visibility if published. Congratulations to the team on this excellent piece of science.

Author Response

-Reviewer 2

0. In my view, an interesting and representative paper to the understanding of bioactive peptides derived from maize. I have no concerns whatsoever on the experimental design nor in the results obtained by the researchers.

[Thank you very much for your encouragement!]

  1. My main suggestion is to ask a professional Editing service to proofread the manuscript to remove typographical errors and/or loose, unstructured sentences. There are some questionable constructions (e.g. what does College represent to the version of Minitab? line 473; Suiza at line 430, which I assume the authors intended to mean Switzerland?).

[Than you for your suggestion, however this MS was previously revised and a certificate of English editing is attached to this letter. However we modified now at line 478, State College, USA is the municipality where the Minitab, LLC’s headquarter is located; and now line 435 the word Suiza has been changed to Switzerland].

  1. For instance, if one considers the abstract alone, a few suggestions that could be made are:

   a)Please remove sentences from line 15 up to the word ‘scarce’ – line 17 on the  abstract.

[The sentences have been removed.]

    b) Substitute penetrating (line 22) to penetration

[At line 22, the word has been changed accordingly]

    c)Remove ‘a’ from line 29

[At line 29, the word “a” has been removed.]

   d)Apart from that, please remove the table from Figure 7, which is now bound to the same figure file. I suppose tables should not be sent as figures.

[The table has been removed from figure 7 and now it is presented as Table 2 at line 213.]

  1. These suggestions do not take merit away from the paper, but indeed could help it gain some more visibility if published. Congratulations to the team on this excellent piece of science.

[Thank you very much for your encouragement!]

Round 2

Reviewer 1 Report

After re-reviewing the manuscript, I accept it as presented.

Author Response

-Reviewer 1

  1. After re-reviewing the manuscript, I accept it as presented.
    [Agree: Thank you very much for your observations.]